# Pneumonia and Invasive Pneumococcal Diseases: The Role of Pneumococcal Conjugate Vaccine in the Era of Multi-Drug Resistance

**DOI:** 10.3390/vaccines9050420

**Published:** 2021-04-22

**Authors:** Chiara Scelfo, Francesco Menzella, Matteo Fontana, Giulia Ghidoni, Carla Galeone, Nicola Cosimo Facciolongo

**Affiliations:** Pneumology Unit, Department of Medical Specialties, Azienda Unità Sanitaria Locale-IRCCS di Reggio Emilia, 42100 Reggio Emilia, Italy; francesco.menzella@ausl.re.it (F.M.); matteo.fontana@ausl.re.it (M.F.); giulia.ghidoni@ausl.re.it (G.G.); carla.galeone@ausl.re.it (C.G.); nicola.facciolongo@ausl.re.it (N.C.F.)

**Keywords:** pneumococcal colonization, pneumonia, vaccines, anticapsular antibodies, humoral immunity, serotypes

## Abstract

*Streptococcus pneumoniae* related diseases are a leading cause of morbidity and mortality, especially in children and in the elderly population. It is transmitted to other individuals through droplets and it can spread to other parts of the human host, causing a wide spectrum of clinical syndromes, affecting between 10 and 100 cases per 100,000 people in Europe and the USA. In order to reduce morbidity and mortality caused by this agent, pneumococcal vaccines have been developed over the years and have shown incredible effectiveness in reducing the spread of this bacterium and the development of related diseases, obtaining a significant reduction in mortality, especially in developing countries. However, considerable problems are emerging mainly due to the replacement phenomenon, multi-drug resistance, and the high production costs of conjugated vaccines. There is still a debate about the indications given by various countries to different age groups; this is one of the reasons for the diffusion of different serotypes. To cope with these problems, significant efforts have been made in the research field to further improve vaccination serotypes coverage. On the other hand, an equally important commitment by health care systems to all age group populations is needed to improve vaccination coverage.

## 1. Introduction

*Streptococcus pneumoniae* (SP) is a bacterium responsible for noninvasive diseases such as non-bacteremic pneumonia, otitis media, and rhinosinusitis but it can also cause invasive diseases like bacteremic pneumonia, sepsis, meningitis, or invasive pneumococcal disease (IPD). Nowadays, it remains a leading cause of morbidity and mortality, especially in children and the elderly population. Indeed, in 2005, the World Health Organization (WHO) estimated that 1.6 million deaths were caused by this agent annually [1]; in particular, a very significant fact is that globally, 5% of all cause-child mortality under five years old were due to pneumococcal infections (last update on March 2012) [2]. In Europe and the United States (US), SP appears to be the most common cause of community-acquired bacterial pneumonia (CAP) in adults. In these regions, the annual incidence of IPD ranges from 10 to 100 cases per 100,000 people [1]. In Europe, the incidence of CAP varies by country, age, and gender, it increases sharply with age, and it is more common in men than in women [3]. For example, in Spain, the EVAN-65 study found an incidence rate of overall CAP of 14 cases per 1000 person/year, with an increase by age and male gender [4]. In 2010, 26 European countries reported 21,565 cases of IPD to The European Surveillance System with an overall incidence of 5.2 cases per 100,000 population, with the most affected age groups being <1 year and ≥65 years old [5]. In Italy, in 2019, 1671 cases of invasive pneumococcal diseases were reported [6].

The aim of this narrative review is to provide an overview on SP and the clinical features of SP related diseases; moreover, we want to highlight the emerging problem of the reduction in available effective therapies and therefore the role of the pneumococcal conjugate vaccine in this context, its characteristics, and its effect regarding pneumonia and IPD and to give an overview on alternative strategies to face the spread of pneumococcal diseases.

### 1.1. Streptococcus Pneumoniae Serotypes and the Mechanism of Infection

SP bacteria are lancet-shaped, Gram-positive, facultative anaerobic organisms. The pathogenicity comes from capsular polysaccharides, which are antigenic and form the basis for classifying pneumococci by serotypes. The reservoir for pneumococci is most likely the nasopharynx of asymptomatic human carriers, it is possible to find them in 5% to 90% of healthy persons [7]. Direct person-to-person contact via respiratory droplets or autoinoculation in persons carrying the bacteria in their upper respiratory tract are the way of transmission of SP [7].

Over the years, 100 different serotypes of this human pathogen have been discovered [8,9]; among them, 23 serotypes are responsible for 80–90% of invasive infections, with some serotypes (1, 2, 4, 5, 7F, 8, 9,12F, 14, 16, 18C, 19A) more invasive compared to others (3, 6A, 6B, 11A, 15B/C,19, and 23F) [10]; specifically, serotypes 1 and 3 are more often isolated from patients with pneumonia, whilst 6, 10, and 23 are usually isolated from patients with meningitis and serotype 14 and 19A are the most common worldwide for IPD [11]. However, 62% of invasive diseases worldwide is due to ten most common types [11].

Colonization is the first necessary step for the development of the disease. This is performed by the polysaccharide capsule which enables SP to evade entrapment by mucus in the respiratory tract and at the same time it defends SP from opsonophagocytosis and detection by host receptors. These characteristics lead SP to survive the early innate defense mechanisms of the respiratory tract [12]. Then, the epithelial damage and ciliary slowing is carried on by toxins like pneumolysin, hyaluronidase, and the production of hydrogen peroxide (H_2_O_2_). At the same time, the attachment of SP to the epithelium is initially made by non-protein virulence factor (phosphorylcholine), then reinforced by protein adhesins (pneumococcal surface proteins (Psp) A and C and lipoprotein pneumococcal surface adhesion (Psa)A [12] and for some serotypes also by pilus-like structures. Capsular serotypes also influence tissue invasion and the pneumococcal surface protein expression. The next step in the etiological pathway is to reject host immune defenses and antibacterials, producing a biofilm which isolates the bacterium and grants it the ability to conceal and resurface when the host’s defenses drop. The following steps mostly depend on interactions between the pathogen and the host’s defenses and the expression of higher levels of some proteins. The entry of the pathogen into the bloodstream and the eventually ability to invade the central nervous system is allowed by the interaction between the epithelial polymeric immunoglobulin (Ig) receptor (that normally transports secretory IgA and E-caderin) and the PspA/PspC [12]. Indeed, in severe pneumococcal pneumonia, the cytotoxic and proinflammatory effect of pneumolysin produces lung injury and epithelial damage, leading to translocation of SP from the alveoli to the interstitium and finally the bloodstream [12].

Host defenses could fail at several levels: the first lane defense is cough reflex, mucous, mucociliary escalator, and humoral factors. The critical point is the cell-mediated host defense, initially born by macrophages and, in the event of their failure, continued by neutrophils activated by chemokines [13]. The neutrophilic action is important mainly in restricting the infection but it could also be responsible for lung injury [13].

### 1.2. Pneumococcal Vaccines Overview

In order to reduce morbidity and mortality caused by this agent, pneumococcal vaccines have been developed over the years. Basically, nowadays there are two categories of vaccines available: pneumococcal polysaccharide vaccine (PPV) and pneumococcal conjugate vaccine (PCV). The first is a preparation containing 23 serotypes (PPV23), based on capsular polysaccharides, and it was licenced in 1983 in the US, replacing the first pneumococcal vaccine approved in 1977 with 14 serotypes [7,14]. One dose of PPV23 is currently recommended by the Centers for Disease Control and Prevention (CDC) for adults aged >65 years in the US, eventually preceded by a dose of PCV13 one year before in case of increased risk for exposure to PCV13 serotypes; the combination is as well recommended for adults aged ≥19 years with immunocompromising condition, CSF leak, cochlear implant [14]. In the past, it became evident that the polysaccharide vaccine had various limitations, such as poor immunogenicity in children, an inability to generate immune memory, and no effect on pneumococcal carriage. For those reasons, a different kind of vaccine was developed, the conjugate one, in which capsular polysaccharides were conjugated with a carrier protein, overcoming most of the problems of the polysaccharide vaccine. The carrier protein induces a T-cell dependent immune response, consisting in immunologic memory and antibody response in adults and young children [15]. The first licensed conjugated vaccine was heptavalent (PCV7, Prevnar 7^®^), licensed in the US in 2000, then preparations with more serotypes were developed, in particular 10 (PCV10, Synflorix^®^) and 13 (PCV13, Prevnar^®^) (Table 1) [16]; they were both registered in Europe in 2009, whilst the latter was licenced in the US in 2010, firstly only for children, then also in adults [7,15]. PCVs are administered in the first year of life with two or three doses given in the first semester of life and with a booster dose at about one year in several countries. This practice had completely modified the total burden of pneumococcal diseases in vaccinated children and their unvaccinated contacts of any age [17]. Nevertheless, they have various limitations that reduce further improvement: they cannot be easily included in the national immunization schedule of the poorest countries because of their complexity and their high expansive cost of preparation [16].

As for PCVs, it was seen that routine PCV7 vaccination had a major impact on the incidence of invasive and noninvasive pneumococcal diseases in children worldwide; however, the prevalence of some non-PCV7 serotypes increased. The introduction of pneumococcal vaccines containing a wider range of serotypes was the solution to broader serotype coverage, likely with a positive impact on decreasing serious infections [18].

In Italy, the vaccination plan contemplates the use of PCV for children in their first year of age (three doses in total), a scheme of a first dose of PCV followed by a dose of PPV23 at least two months later for older children, adolescents, adults aged 19–64 with risk factors, and all adults over 65 years [19]. In the United Kingdom, the vaccination plan changed in 2020 for children with only 2 doses of PCV instead of three, at 12 weeks and 1 year; besides, people aged 65 and over only need a single pneumococcal vaccination (PPV), as well as people at high risk for long-term health conditions, with an eventual repetition every five years [20].

On the international scene in 2014, after the results of the CAPiTA study [21], the Advisory Committee on Immunization Practices (ACIP) recommended, for adults over 65 years who have never received a pneumococcal vaccine before, a first dose of PCV13 followed after 6 to 12 months later by a dose of PPV23 [15], but in 2019, there was a rectification for immunocompetent adults [22] (Table 1). The ACIP recommendation for immunocompetent children [23], immunocompromised children [24], and immunocompromised adults [25] had already been expressed in 2012 and 2013.

### 1.3. Overview on Serotype Independent Pneumococcal Vaccine

There is a need for a serotype independent vaccine derived from the emerging problem of serotype replacement and also from the emerging non-encapsuleted SP (NESp), causing both invasive and non-invasive diseases. Moreover, considering the number of serotypes discovered, a vaccine that can cover the majority of serotypes is unthinkable. The success of the research in this field mostly depends on availability of appropriate animal models that represent characteristics of humans’ diseases. The search is now focused on conserved surface-exposed and immunogenic protein antigens, with the potential to induce broad protection and to prevent serotype replacement. Several proteins have been investigated in animal models [26] and also in human models [27]: surface proteins (such as choline binding protein PcpA, PspA PspC, hydrolases, pore-forming toxin pneumolysin), a lipoprotein protective against pneumococcal carriage (lipoprotein PsaA), D-alani-dalanin-cardoxypeptidase B DacB, methionine binding protein Q MetQ, purine nucleoside receptor A PnrA, protein required for cell separation in group B streptococci PcsB, pneumococcal peptide 27 (Pep27), and elongation factor Tu (EF-Tu), a surface-accessible protein found in the bacterial cytoplasm and culture supernatant. All these proteins are involved in the immune response and in some step of bacterial etiological processes [28]. None of these studies reached the final goal, so there is increasing interest in combined formulations including different proteins and delivery systems, which may impair different stages of pneumococcal infection, potentially providing large protection (for example, PspA and pneumolysin combined or with other antigens). There is also interest in studying this new formulation in combination with PCV10 and PCV 13 [29].

To overcome the difficulties faced with this approach, killed whole cells or live-attenuated bacteria emerged as an alternative option. This vaccine, presenting the antigen in their natural conformation on the bacterial surface, induces the production of protective antibodies and cellular immune response (CD4+ T cell response), without requiring purification of individual proteins [11]. The whole-cell vaccine (WCV) has some limitations too: the live-attenuated one gives concern for the risk of reactivation of the pathogenicity, while the inactivated WCV needs multiple dosages and it does not confer long-term humoral immunity [11].

More opportunities are emerging from SP genomic sequencing, which, for example, allowed to discover the pneumococcal histidine triad protein and E (Pht) which demonstrated to be protective by inducing strong Th17 responses in animals [28].

## 2. Clinical Features of Invasive and Non-Invasive Infections

SP usually colonizes the nasopharynx, which is the real reservoir of the pathogen, then it is transmitted to other individuals through droplets and it can spread to other parts of the human host, causing a wide spectrum of clinical syndromes [30,31]. Pneumococci can spread directly through the upper respiratory tract, causing otitis and sinusitis or through the lower respiratory tract, causing pneumonia [13].

In children, otitis media is a common consequence of nasopharynx colonization, together with other mucosal affections such as sinusitis [30]. The most severe consequences of pneumococcal infections are due to the epithelial cell surface penetration of the bacteria, known to be hematogenous spreading (bacteremia), meningitis, and invasion of the pleural space via the respiratory epithelium (empyema); these three manifestations, hitting sites that are normally sterile, define the so-called IPD, which is known to have a high mortality rate, particularly in immunosuppressed patients [13,32].

### 2.1. Non-Invasive Infections

Focusing on non-invasive infections, the main manifestations are sinusitis, acute otitis media, and pneumonia. The main symptoms of otitis are ear pain, discharge, and fever [33], while sinusitis is characterized by discharge, cough, dyspnea, and facial pain [34]. Finally, pneumonia is characterized by cough, mucus production, fever, tachypnoea, dyspnea, chest pain, and respiratory failure; incubation period for pneumonia is about 1 to 3 days [30]. The burden of pneumococcal disease in children is mainly represented by sinusitis and otitis, in particular, acute otitis media accounted for about 5 million cases in the pre-vaccination era in children [30]. In adults, pneumonia represents the main burden of the disease, with sinusitis and otitis having a small impact on the epidemiology [35]. Pneumonia in children is estimated to cause more than 600,000 deaths worldwide, in particular, in developing countries, causing an emerging health problem [36]. In children aged between four months and five years, SP is the main etiological agent causing pneumonia, but it remains the second cause also in adolescents and older children [37]. In adults, SP is the leading cause of CAP and its incidence in adults has been estimated between 11 and 23 cases per 100,000 population before the spreading of the vaccination, with a wide range of mortality, ranging from 6 to 40%, depending on the setting and the clinical features of patients [32]. In particular, factors associated with short term mortality are diabetes, male gender, neurological disorders, malignancies, bacteremia, tachypnoea, multilobar infiltrates, pleuritic chest pain, hypotension, hypothermia, and leucopenia [35]. Scores and criteria have been produced and validated over the years in order to identify the risk factors associated with severe pneumonia, the access to intensive care unit, and the consequently high mortality rate [38]. It is also important to say that the long-term mortality after pneumonia is about 50% and it is associated with the presence of malignancies, chronic obstructive pulmonary disease, and cardiovascular diseases [35]. Influenza is a known risk factor for the development of pneumonia and it has been demonstrated that the simultaneous vaccination for SP and influenza has a synergic effect [30,32].

### 2.2. Invasive Pneumococcal Disease

The transition to IPD is a complex pathologic process that involves many different features: the serotype and the consequent pathogenicity of the microorganism (given by the capsule and the protein pneumolysin), the mucosal features, the immune response of the host (represented primarily by the T cells activation and the action of macrophages), and pre-existing clinical conditions [12,13]. Serotype 3 is the most concerning because it is associated with high mortality rate [39], worst respiratory failure [40], and septic shock [41], compared to other serotypes. In European countries, the main predominant serotypes are nowadays 15 B/C, 11A, 23B, 15A, 35B, 10A, 21, and 23A, with relative prevalence of some serotypes compared to others in different countries [42].

IPD incidence at the beginning of the conjugate vaccine era was 23.2 cases per 100,000 in the United States, with children younger than 2 years and elderly people being more at risk [43]. The incidence is highly variable and depends on factors such as age, socioeconomic level, immune status, and geographic area [12]. As previously stated, bacteremia, meningitis and pleural infection are the hallmarks of IPD, and bacteremia without a focus of infection is the most common presentation in children aged 0–5 years [44]. In children, bacteremia accounts for about 70% of IPD, while SP is the leading cause of meningitis among children aging 5 years or younger; functional or anatomical asplenia, cochlear implants, immunodeficiencies (such as human immunodeficiency virus-HIV), and some ethnicities are associated with a higher risk of developing IPD [30,45].

More than 100,000 cases and more than 60,000 deaths worldwide in children younger than 5 years of age are estimated due to pneumococcal meningitis [24]. In adults, pneumococcal meningitis accounts for about 50% of all causes of bacterial meningitis, involving between 3000 and 6000 cases per year in the US [30]; meningitis is considered to involve approximately 5% of all cases of IPD [35]. Some conditions are associated with a greater risk of evolution to IPD: in immunocompetent hosts asthma, chronic lung, kidney and heart diseases, alcohol abuse, cigarette smoking and diabetes, while in immunocompromised hosts, HIV infection, anatomical or functional asplenia, solid or hematological malignancies, solid organ or hematopoietic cells transplantations, sickle cells disease [31]. Risk factors for mortality in IPD have been recently evaluated and identified. In particular, male sex, age of more than 45 years, drug or alcohol abuse, smoking habit, previous tuberculosis, infection with PCV13 serotypes, penicillin resistance, presence of comorbidities such as cardiovascular, chronic kidney disease or dialysis, chronic liver disease, neurological diseases, or high Charlson comorbidity index (≥2) have been identified as conditions associated with increased mortality. On the other hand, during the course of the disease, septic shock, respiratory failure, mechanical ventilation, intensive care admission, use of vasopressors, and hospital-acquired infections have been associated with a higher mortality [39,46].

## 3. Therapeutic Option

Pneumococcal diseases are primary treated with antibiotic therapy, but in recent years, the selective pressure due to the overuse of these drugs against SP led to the development of some resistances against the most used antibiotics.

### 3.1. Pneumonia Treatment

Focusing on pneumonia, the latest guidelines recommend, considering SP a leading cause of CAP but also the changing epidemiology of these infections, the use of an empiric therapy active on SP and other common pathogens: amoxicillin, doxycycline or macrolides for healthy patients, and combination therapy with amoxicillin/clavulanate or cephalosporin and a macrolide or doxycycline, or a monotherapy with a fluoroquinolone in patients with comorbidities. The use of other drugs (such as vancomycin) should be considered only in severe cases non-responding to first- and second-line therapies, since another worrying element is the emergence of vancomycin-tolerant strains [47]. Another crucial point is the early administration of antimicrobial therapy [48].

### 3.2. Drug Resistance

Resistance to antibiotic treatments has been affecting many classes of drugs since the end of the 1960s, but the main problem regards resistance to β-lactams and macrolides; however, resistance mechanisms for almost all antibiotic families used for the treatment of SP (macrolides and azalides, penicillin and β-lactam resistance, fluorochinolone, tetracycline, clindamycin) have been documented over the last 25 years, to a lesser extent for cephalosporin (cefotaxime and ceftriaxone) [12,49,50]. The circulation of multidrug resistance (MDR) SP, which is defined as resistance to ≥3 antibiotic classes (usually β-lactam, macrolides, tetracycline, and sulfonamides), is increasing worldwide and represents more than 30% of pneumococci [51]. Patients developing pneumococcal diseases are more likely to have antimicrobial resistance if they were previously exposed to antibiotics, recent hospitalization, young age (<2 years), and lack of breast feeding; if they have immunosuppressing underlying conditions; and if they attend daycare centers [52]; nevertheless, the time elapsed from the most recent treatment and a new infection is more significant than cumulative prior antibiotic exposure to predict the development of pneumococcal resistance. It declines rapidly in the first month after antibiotic exposure for penicillins, cephalosporins, and fluoroquinolones, whilst the decline in resistance appears to be slower for macrolides [53].

Antimicrobial resistance has also an important economic impact in the management of pneumococcal diseases [32] and will probably modify the vaccination strategy as well, based on the emerging serotypes in response to the antimicrobial pressure [54]. The resistance to penicillin is determined by the modifications of the penicillin-binding proteins within the cell wall; it has a prevalence up to 50% of all pneumococcal infections in some countries such as Spain [12]. The resistance to macrolides, which also has a high prevalence of up to 50% in countries like France and Italy, is determined by various mechanisms, the most important of them regarding the presence of inactivating enzymes or efflux pumps [49]. The resistance to fluoroquinolones is determined by more than one mutation of genetic loci for topoisomerase IV or DNA gyrase, but is uncommon, usually less than 2% [12]. Then, the important point is how to treat SP infections with a known resistance to the most commonly used antibiotic classes. Some strategies are suggested: for penicillin resistance, higher dosages or shifting to third generations cephalosporins or fluoroquinolones; in high macrolides resistance areas, a combination therapy (e.g., with a β-lactam) appears to be the best choice; finally, even if fluoroquinolones resistance is an uncommon event, these drugs are recommended to be given at the appropriate dosage in order to avoid the occurrence of more than one mutation [12].

The resistance spectrum depends also on the geographic area, and therapeutic strategies need to be titrated following this paradigm. In the recent CAP guidelines, for example, the use of a single therapy with macrolides in patients without comorbidities or risk factors for *Pseudomonas aeruginosa* or methicillin-resistant *Staphylococcus aureus* (MRSA), is subordinated to a low documented pneumococcal macrolide-resistance (<25%) in the specific area [48]. The introduction of vaccines has changed the antimicrobial resistance profile of SP nowadays, but there is a great geographical variability: penicillin resistance has been observed in particular in South Africa, Far East, and Middle East (>50%) while in European countries, is lower (15%); macrolide resistance is high in Far East, South Africa, and southern Europe, while is low in northern Europe and Australia; combined resistance is significant in Spain; quinolone resistance is significant in Poland, Spain, Italy, in several Asian countries, and in Canada [55]. Moreover, the isolation of different serotypes has a role for the presence of antimicrobial resistance [56]. In terms of therapeutic options, a combination therapy is widely considered the best strategy regarding widespread resistance profiles [12].

Nowadays, we need strategies to limit the development of antimicrobial resistance; diminishing the need for antibiotics by decreasing the incidence of pneumonia is a first step. That is why a strategy of expanded vaccination targeting SP is a way to decrease antimicrobial resistance, as it reduced the overall prevalence of invasive bacterial diseases (i.e., pneumonia, meningitis, and sepsis). After introduction of the pneumococcal conjugate vaccine, the estimated number of cases of IPD in all age groups, caused by strains with reduced susceptibility to penicillin or multiple antibiotics, decreased by half. Therefore, pneumococcal conjugate vaccine is one strategy to reduce the burden of antibiotic resistant bacterial diseases globally [57].

### 3.3. Other Therapeutic Strategies

Regarding other therapeutic strategies, the management of SP pneumonia could be in an out-patient or in an in-patient setting, depending on the clinical features of patients [48]. Supportive therapies are important, in particular, in severe infections: when respiratory failure occurs, respiratory support from oxygen therapy until mechanical ventilation in the event of acute respiratory distress syndrome (ARDS) [58] could be necessary. No recommendations are made for non-invasive ventilation in this setting [59], while other emerging noninvasive strategies are chosen in order to avoid the use of mechanical ventilation, such as high flow oxygen therapy (HFOT) [60]. Finally, steroids are a very discussed therapeutic option for pneumonia, in particular, in severe ones [61]; recent evidence suggests that they could have a role in the mechanically ventilated patient with moderate to severe ARDS [62].

The search for novel treatment strategies for SP infections is ongoing. It would be useful to identify potential new drug targets and to improve infection outcomes; the strategies to achieve this goal, each with benefits and limitations, could be boosting host immunity by interfering with its immune responses, lowering pneumococcal virulence, and developing novel antibacterial with a new mechanism of action. More research is needed in this field [63].

## 4. Immunological Features: The Immune Response to the Vaccine

Multivalent PPV have been available for many years now. The PPV23 is recommended by public health agencies around the world for adults over the age of 65 and other high-risk individuals. However, its use involves some warnings because the immunological response to PPV23 tends to decrease over time, with a consequent decrease in efficacy and protection [64,65]. PCV tend to be more consistently immunogenic than PPV and give a prolonged memory duration of the pneumococcal immune response [66]. The crucial point is that vaccination with PPV produces an immune response determined only by the stimulation of B lymphocytes which become activated and then, as plasma cells, produce antibodies [66,67]. PPV23 is known to induce a T-independent (TI), exclusively humoral, immune response without the ability to establish immunological memory. Moreover, PPV23 could not elicit secretory IgA in the nasopharynx [11]. Polysaccharide antigens, such as pneumococcal antigens contained in PPV23, stimulate pre-existing memory B cells (MBC) towards terminal differentiation into antibody-producing cells, thus leading to the overall depletion of the MBC pool (Figure 1) [68]. This depletion of the MBC pool, driven by PPV23, has been demonstrated in a small number of studies enumerating MBC after immunization with PPV23. The accumulated data from these studies showed that vaccine-naïve adults had pre-existing serotype-specific pneumococcal MBC existing prior to immunization. This has been attributed to pneumococcal naso-pharyngeal transport and previous pneumococcal infections [69,70]. Significant increases in MBC have been reported after 7 days and one month after immunization with a single dose of PCV among healthy adults [69], adults with various immunocompromising conditions [71], and elderly subjects [72]. It is significant to define the predictive value of MBC for the persistence of humoral immunity to establish how much MBC correlates clinically with protection. To date, antibody levels present after vaccination have been considered and have been widely used for this purpose, but it is likely that MBC can better predict long-term protection than current serological measures [68].

Unlike PPV, the distinctive feature of PCV is the presence of pneumococcal polysaccharide antigens covalently linked to an immunogenic carrier protein; thus, the carrier protein induces a humoral immune response strictly dependent on T lymphocytes, which induces B cells to produce antibodies and generate immune memory through b cell memory and long-lived plasma cells (Figure 2) [73,74]. The carrier protein is processed by polysaccharide-specific B cell, and the peptides are presented to the carrier peptide-specific T lymphocytes, resulting in the aid of the T lymphocytes for production of plasma cells and MBC [75]. The persistence of serum antibody levels above the protective threshold is the critical point of the immune response after immunization with PCV, and there is evidence that these vaccines induce the production of anti-gen-specific MBC [73,76].

Ongoing studies examining the response of MBC to pneumococcal vaccination will be extremely important to our understanding of the long-term protection induced by PCV.

All these characteristics determine an increase in the power and duration of the immune response, even towards subsequent exposures to other vaccines-types (VT) with different pneumococcal strains. In this regard, some studies evaluated and compared the immune response to PCV and PPV in immunocompetent or elderly healthy adults, and the antibody response was usually superior with PCV [75,77]. A sustained and significant immune response was demonstrated in HIV positive individuals as well, for up to 5 years [78]. PCV vaccines are recommended for all children and showed to prevent nasopharyngeal transport of pneumococcal strains and to protect patients from IPD, pneumonia, and otitis media [79,80,81]. Childhood vaccination programs including PCV demonstrated a significant reduction in pneumococcal diseases in adults as well as observed thanks to the herd effect [82,83]. CAPiTA was a large, prospective, randomized, placebo-controlled clinical trial conducted to evaluate the clinical efficacy of PCV13 in the elderly. This trial involved over 84,000 participants >65 years of age [21]. Vaccine efficacy was 30.6% against pneumococcal CAP and 75% against VT-IPD. Furthermore, the magnitude and duration of humoral immune responses in adult at-risk subjects assessed by opsonophagocytic tests and the geometric mean antibody titers were comparable to those of the overall population and persisted for at least 2 years [84]. A post hoc analysis predicted that vaccine efficacy could drop from 65% for 65-year-old adults to 40% for 75-year-old adults. This aspect indicates that patients should be vaccinated as soon as possible to avoid the decline in efficacy due to immunosenescence [85]. The 7-valent pneumococcus diphtheria-conjugated polysaccharide vaccine (PCV7) induces a powerful immune response in children and reduces nasopharyngeal transport of VT, recurrence of acute otitis, and the number of invasive disease episodes [86]. Currently, this vaccine is not yet authorized for use in adults, although preliminary clinical studies showed that in healthy subjects over 70 years of age, PCV7 induced a greater functional antibody activity after vaccination than PPV23, even though it showed a reduced response in previously vaccinated patients [66,87]. One randomized control trial (RCT) enrolled and randomized 150 patients with chronic obstructive pulmonary disease (COPD) to PPV23 or PCV7. IgG production was higher in the PCV7 group than in the PPS23 group for all seven serotypes, reaching high statistical significance for five of these. PCV7 resulted in a higher kill opsonization index (OPK) for six out of seven serotypes. In this study, one month after vaccination, PCV7 showed to induce an immune response greater than PPV23 in COPD, with lower efficacy rates in elderly subjects and previous PPV23 vaccinated [88]. An interesting and recent study evaluated the immunogenicity and safety of PCV13 in 74 PPV23 naïve patients and 58 previously immunized PPV23 patients (>1 year ago) with severe (stage 4–5) chronic kidney disease (CKD) [89]. The authors quantified specific serum IgG, IgM, and IgA for seven serotypes, at baseline, 4 weeks, and one year after vaccination. Baseline concentrations for most serotype specific IgG and IgM and serotype 3 specific IgA were higher in subjects previously immunized with PPV23 than in PPV23 naïve patients. PCV13 significantly increased almost all specific immunoglobulins. The increase in antibody concentrations and the proportion of patients with >2-fold increase after immunization were generally greater in PPV23-naïve subjects than in previously immunized patients. Data from this study therefore demonstrated that CKD patients who received previous PPV23 immunization more than a year earlier had an antibody response to PCV13 lower than in naïve subjects [90].

A very recent RCT performed in Gambia compared the immunogenicity and safety of a new ten-valent PCV (SIIPL-PCV) containing serotypes 1, 5, 6A, 6B, 7F, 9V, 14, 19A, 19F, and 23F with the PCV13 vaccine [89]. The immunogenicity of SIIPL-PCV was lower than that of PCV13, for which there are numerous data on efficacy and safety against pneumococcal disease.

## 5. Medical Costs

Regarding the important economic aspects, all the highly significant studies available in the literature confirm that pneumococcal vaccinations are cost-effective in children, adults, and in elderly patients [91,92]. The great clinical impact of pneumococcal immunization is now evident in adults and in the elderly, and pharmaco-economic studies and literature reviews support and confirm the cost-effective profile of pneumococcal vaccination in all ages [93,94]. Although the PPV23 vaccination is usually convenient, further scientific evidence should be considered in the decision-making process prior to implementation, and this due to the limited duration of immunological protection of this vaccine. PCV13 vaccination is also affordable, although pharmaco-economic studies need updating after the publication of the CAPiTA trial data [11]. Vaccination programs will become increasingly important around the world in the future due to the constant aging of the population, and prevention may reduce the burden of pneumococcal diseases on health systems and the consequent incremental social costs. In light of these considerations, pneumococcal vaccines are a paradigmatic example of a clear and effective trade-off between benefits and costs [95,96]. PCV13 and PPV23 have an excellent value for money and should become a real priority for decision makers in public health [97]. One of the principal benefits deriving from the introduction of vaccines has been the reduction in hospitalization due both to invasive and non-invasive pneumococcal infections and the absolute reduction of hospitalizations for all-cause pneumonia in children in the US and Europe [9,95,98]. The positive effect on reduction in hospitalization and mortality becomes more significant considering the comorbid population and the subgroup of immunocompromised patients [44].

Vaccination with PCV is expensive and the efficacy in adults is difficult to establish, given the effect on prevalence coming from vaccinations in children [99]. The pharmaco-economic aspects represent a challenge in each country in evaluating which vaccine to recommend in each population group. José and Brown underlined two interesting aspects that could increase cost-effectiveness of PCV in adults. First, the reduction of COPD exacerbations gained with PCV in adults, although further data are needed on the efficacy of PCV against all infective exacerbation of COPD. Second, the possibility of altering the PCV with serotypes that cause lung infections in adults, given that PCV 13 was born to cover serotypes responsible for children infections. To the first point, it is important to add that a great benefit would be obtained considering the whole respiratory population at risk, such as asthmatics and patients with primary or secondary interstitial lung diseases [100,101]. The discussion becomes even more difficult when we consider that the population to be vaccinated is elderly. In fact, in this age group, the immune response to both SP infections and vaccines is influenced by the chronic low-level inflammation, known as “inflammaging” [99]. Novel strategies are needed to extend the coverage in the elderly population.

## 6. Discussion

It is now known that the spread of pneumococcal vaccines has reduced the prevalence of invasive and non-invasive pneumococcal diseases over the years. However, the protection conferred by vaccines against more pathological serotypes produced the emergence of other serotypes with renewed pathogenicity; this is increasingly emerging from epidemiological data and it is consistent with the recent scientific literature. Routine use of PCV in infants leads to an unexpected reduction in nasopharyngeal colonization with vaccine serotypes of SP [1]. As children are the main reservoir and transmitters for SP diseases in adults, the reduction in colonization serotypes is an important factor in increasing herd immunity. However, this success is limited by the fact that 20% of pneumonia cases are still caused by PCV13 serotypes and by the expansion of non-vaccine serotypes [1]. In this context, the examination of changes in pneumococcal nasopharyngeal carriage after introduction of PCV13 could be of considerable interest to predict circulating serotypes despite vaccines or to discover the presence of serotypes with renewed pathogenicity.

Benefits from vaccination in preventing pneumococcal diseases in young children are well established, reaching a reduction of 57% and 75% in pneumococcal related-deaths respectively in immunocompetent children and in children with HIV infection [28,102]; on the other hand, the effectiveness in adults was only evaluated in two randomized control trials: one is the CAPiTA study [21] and the other one is among HIV patients [103]. The CAPiTA was the first trial to demonstrate the vaccine efficacy against vaccines serotypes for CAP, but it was conducted in a setting with an existing infant pneumococcal vaccination program with PCV. The efficacy may vary in the population that received other vaccine types or depending on epidemiological spread of serotypes in different countries or on population susceptibility to pneumococcal infections. The second study was conducted in a particular immunocompromised population but the efficacy was demonstrated in this subpopulation as well [103]. Finally, the effectiveness in protection from non IPD pneumococcal CAP has been demonstrated only for PCV13 [15].

PCV have shown numerous benefits over time but also some defects, such as the emergence of non-vaccines serotypes, vaccine-induced replacement serotypes, protection against some of the included serotypes lower than desired, and perhaps inadequate coverage in the elderly; for these reasons, studies on bacterial proteins, which could provide serotype independent protection, are still ongoing, but nowadays they have not reached phase III yet [104,105]. There are, in particular, three reviews which give a comprehensive overview of the vaccines under study [106,107,108], but last year, Masonian et al. made a precise update with a full list of ongoing clinical trials [11].

The replacement phenomenon is one of the major emerging trouble. This problem has developed since the first conjugate vaccine (PCV7) appeared. In particular, serotypes 1, 3, 5, 6A, 7F, and 19A were the most responsible for new invasive infection, so they were added to serotypes included in PCV7 (4, 6B, 9V, 14, 18C, 19F, 23F) creating PCV13. Over the years, comparative studies about efficacy between different vaccines have followed, resulting in no substantial difference in the reduction of the incidence of IPD, as demonstrated by McGirr et al. [109]. The discussion grows if we consider the comparison between the serotype coverage offered by PCV13 and PPV23. In fact, although the PCV has fewer serotypes than 23-valent PPV, its immunogenicity has the power to increase and promote “herd immunity”, reducing the rate of asymptomatic carriage and IPD both in children and in the elderly [110]. In their article conducted on a large population from Europe, Americas, Australia, and Africa, Vadlamudi et al. highlighted that the introduction of PCV13 produced a significant reduction in IPD burden, all-cause pneumonia and related mortality; in particular, it showed a reduction in mortality from pneumonia of approximately 30% in all adults. In addition, it indicated a reduction in 30-days mortality among adults hospitalized due to pneumonia [111].

In order to provide the best vaccination coverage and avoid the risk of “serotype replacement”, different types of vaccines were studied. Lee et al. described a new 15-valent pneumococcal conjugate vaccine (PCV15) which added two serotypes to the ones included in PCV13 (11A, 22F). These two serotypes were the most prevalent non-vaccine types isolated in 2009–2012. This vaccine was composed with pneumococcal polysaccharides serotypes activated with molecular linker, adipic acid dihydrazide, and conjugated to a carrier protein, cross-reacting material 197. In this study, two different groups of Rabbit sera were immunized by each individual conjugate and control vaccine, samples were taken at two-week intervals and were analyzed to measure antibody and functional antibody levels by enzyme-linked immunosorbent assay (ELISA) and opsonophagocytic assay (OPA). Both ELISA and OPA demonstrated a similar or higher serotype-specific functional activity for PCV15 and PCV13, except for 3,11A, 23F. These three serotypes were further optimized. This study demonstrated that the two added serotypes had higher immunogenicity when administrated separately, their efficacy in immunogenicity remained to be established when conjugated to other serotypes in a multivalent vaccine. In conclusion, they demonstrated a non-inferiority to the control vaccine [112].

A phase 1 trial on a 20-valent pneumococcal conjugate vaccine in healthy adults was conducted by Thompson et al. This vaccine contains polysaccharides of pneumococcal serotypes 1, 3, 4, 5, 6A, 6B, 7F, 8, 9V, 10A, 11A, 12F, 14, 15B, 18C, 19F, 19A, 22F, 23F, 33F individually conjugated to a nontoxic variant of diphtheria toxin cross-reactive material 197. The control group underwent a licensed tetanus, diphtheria, acellular pertussis combination vaccine (Tdap). This study demonstrated an increase in IgG geometric mean concentrations (GMCs) and OPA geometric mean titers (GMTs) to the 20 vaccine serotypes, proving that vaccination with PCV20 elicited substantial IgG and functional bactericidal immune response. The overall safety profile of PCV20, established with blood collection at screening, on the day before vaccination and during or at five days after vaccination, was similar to Tdap and after PCV13 administration [113]. The potential of these studies regards the possible increase in vaccine coverage to a greater number of serotypes, in particular, the more aggressive ones; therefore, it would require country- and serotype-specific antigens, which it is well known that are continuously changing. These are evident limitations in developing new types of PCV.

Nowadays, there is another emerging concern: the antimicrobial resistance (AMR), complicated by the widespread use of empirical therapy to treat infections, sometimes having a bad impact on clinical outcomes [114]. As pointed out by Cafiero-Fonseca et al., the diffusion of vaccines will reduce the circulation of antibiotic-resistant serotypes, by reducing pneumococcal infections and consequently leading to a decrease in the use of broad-spectrum antibiotics [115]. All data available for now are concerning PCV7, but recent studies showed the same effect for PCV10 and PCV13 [116]. PCV10 and PCV13, containing the majority of recently emerging strains, could reduce the growing problem of AMR. It is evident how vaccination can be beneficial in countless situations: reducing infections during meeting of several people such as sport and religious events, preventing diseases in elderly who care for grandchildren, increasing herd protection [117]. Vaccines can reduce AMR in two ways: the first is by reducing or eliminating organisms and strains carrying resistant genes (this effect may be exerted reducing colonization and through antibody mediated action); a secondary effect is obtained by decreasing the antibiotic usage due to febrile illness (specially for antiviral vaccines) [115].

Another issue is represented by the great variability in vaccine recommendation worldwide, despite the increasing evidence of PCV being effective in children and older adults [116]. Recently, it has been shown that in areas with relatively low vaccination coverage, even previously vaccinated subjects can be re-colonized and be at risk for IPD [118]. Therefore, it is important that the vaccination of young children and infants is extensive: this may lead to a reduction in SP vaccine serotypes circulation. Furthermore, the level of antibodies after pneumococcal vaccination gradually decreases and it correlates with nasopharyngeal colonization; this should influence health authorities of each country both in term of pharmaco-economics and vaccination policies. It is particularly true for the PPV23, which needs a boost after 5 years from previous PPV23 [35], but it exists also for PCV13 [119]. In this context, the discussion on co-administration of influenza and pneumococcal vaccination opens up, demonstrating positive benefits in reducing mortality, as evidenced by Chan et al. in their prospective 12-month cohort study conducted on older residents. [120,121]. This combined effect could also be due to the protective effect obtained by co-vaccination, which prevents the supposed shift from an innocuous colonizer to an invasive pathogen, promoted in the bacterium by the influenza A virus [122].

Finally, there is the controversial issue related to the presence of various type of PCV: in fact, recently, one systematic research [123] and an editorial [119] have compared the effectiveness of PCV10 and PCV13 and both reached the conclusion that it was comparable for both vaccines. Indeed, it appeared that the PCV10 is able to induce a strong antibody production after a single dose [119,123] and that these levels are high enough to allow effective IPD prevention through protection from nasopharyngeal colonization [124]. Moreover, even if the PCV13 is able to offer protection from serotypes 3, 6A, and 19A, a cross-reactivity indicates that the immune response evoked by the PCV10 for serotype 6A is appropriate [124,125]. Lastly, a recent study demonstrated that PCV13 could not protect all the receivers from colonization and infection from serotype 19A [126] and other serotypes (such as 15A in UK) [127], also due to an increasing number of 19A multi-drug resistant serotypes [118]; moreover, serotype 3 in PCV13 seemed to provide an insufficient immune response [98,119]. However, these data are in contrast with what the European Centers of Disease Control and Prevention (ECDC) reported: in countries using PCV13 in pediatric immunization plans, the incidence of serotype 3 diseases decreased by 11% in elderly, compared with an increase of 51% in countries that used PCV10 [128]. Therefore, it is possible that the problem of serotype 3 could also be represented by an antigenic change due to an expansion of an antibiotic resistant variant, which could be present differently in various countries [126]. Serotype 24F, followed by serotype 3, 12F, 19A, and 10A were the most involved in IPD in children <5 years old in the 2017 ECDC annual report, based on data from 29 European countries [129].

It is clear that there has been a reduction in all-cause pneumonia among children after the introduction of PCV, while the impact in adults is still controversial [130,131]. In particular, the elderly are more affected by non-vaccine serotype IPD, so this group seems less protected after the introduction of PCV, as demonstrated by Vestjens et al. [132]; one relevant issue in this matter is that older adults and the elderly account for the largest percentage of hospitalization for IPD. In conclusion, according to several studies [128,133,134] on the role of vaccines in different populations and data from both countries previously using vaccines (such as Netherlands) and others which had not vaccines plan before, the effectiveness of vaccination coverage varies, but only a small benefit was seen after vaccines introduction, due to the increase in non-PCV serotypes.

Vaccines based on pneumococcal proteins could be a possible solution for this problem and several multivalent formulations have been studied, some of these in association with PCV [29]. This formulation could have the benefit of maintaining the herd effect produced by the PCV while promoting wide protection, regardless of serotypes due to the conjugation of pneumococcal capsular polysaccharides with pneumococcal proteins [28]. As an alternative to multivalent formulations, WCV have shown to confer independent immunity, with low production costs, which could be an exceptional alternative to PCV for developing countries [28,135]. These solutions are exciting and promising, but what is well underlined by Converso et al. is probably the core problem: Is it possible to reach the elimination of nasopharyngeal colonization? Aimed at this goal, it is important to consider that new generation serotype-independent vaccines may not be able to reach it. On the other hand, the elimination of nasopharyngeal colonization raises the question on what could happen to microbiome after pneumococcal elimination, opening the possibility of the emergence of new pathogens. An ideal vaccine at this time is one that would allow a reduction of the bacterial load below pathogenetic threshold, by conferring a mucosal immunity (both protective antibody and T-cell-responses), protecting against pneumococcal diseases but not disintegrating pneumococcal colonization [28,106]. Besides, the development of new omics-technologies in association with informatics advances represent a new emerging field of research.

## 7. Conclusions

All these considerations underline that the choice between available vaccines is not easy, considering also that the vaccine coverage depends on vaccination countries’ plans and on humoral responses provided by each kind of vaccine; finally, it also depends on the divergent serotype replacement phenomenon and the increasing diversity in pneumococcal diseases in North America, Europe, and Australia. In particular, this aspect leads to the emergence of serotypes with different pathogenicity in each country and in each age group, reducing the benefit of expanding the vaccine valence [136]. Waiting to see the results of the ongoing research on vaccines not dependent serotypes, it is highly likely that there will be PCV15 and PCV20 on the market, mostly in developing countries, aiming at reducing incidence of pneumococcal infections and the circulation of resistant pathogens. The main problem is the acceptability of costs for PCV for these countries. Therefore, it is with hope that we expect the development of much cheaper WCV and the decrease in production costs for PCV. The field of research on vaccines is in continuous development and renewal; in particular, pneumococcal vaccines, despite already in use and approved for years, have been involved in 170 studies, currently underway or in the process of being closed [137]. Indeed, some studies focus on new types of pneumococcal vaccines, while other studies concentrate on the use of the present vaccines in developing countries where there is not a structured vaccination program. Finally, some ongoing studies are focusing on vaccinations in specific groups of patients, in order to reach a better protection within the whole population to avoid hospitalization and antibiotic use for severe diseases.

The scientific research in this field has highlighted that PCV and PPV alone will not solve the problem of pneumococcal diseases, nor the replacement phenomenon or the circulation of SP AMR. Therefore, research in this area is very dynamic and oriented towards the discovery and application of new possible combinations.

During the last year, after the COVID-19 pandemic outbreak, the development of vaccines able to produce herd immunity and reduce the spread of infections has been one of the major commitments of world research. Furthermore, the experience in dealing with this pandemic has shown us that diminishing the spread of bacterial infections is possible and it reduces risk of mortality due to a disease like COVID-19 as well.

## Figures and Tables

**Figure 1 vaccines-09-00420-f001:**
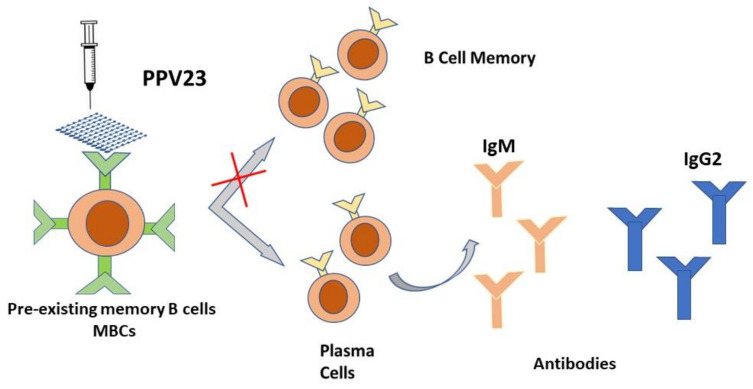
Schematic representation of the T-indipendent mechanism of the PPV 23 vaccine. Polysaccharide antigens, such as pneumococcal antigens contained in PPV23, stimulate pre-existing memory B cells (MBC) towards terminal differentiation into antibody-producing cells.

**Figure 2 vaccines-09-00420-f002:**
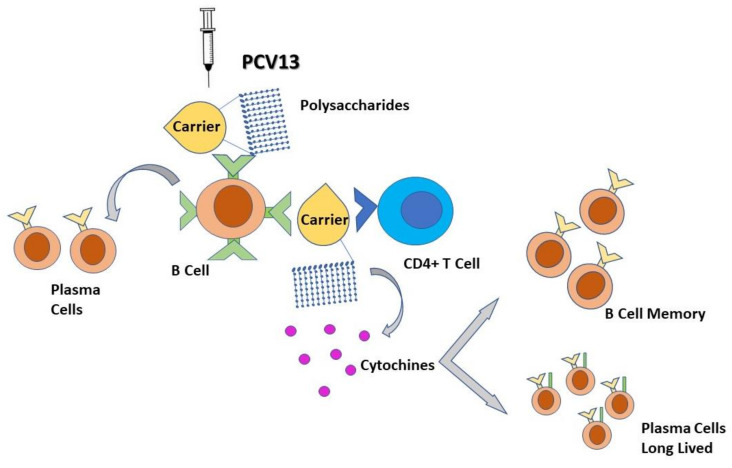
Schematic representation of the T-dependent mechanism of the PCV 13 vaccine. The polysaccharidic portion of PCV stimulates the production of plasma cells, while the carrier portion of PCV activates CD4+ T lymphocytes that induces memory B cell proliferation and long-lived plasma cells.

**Table 1 vaccines-09-00420-t001:** Comparison between commercial vaccines and indications of use.

Vaccine	Serotypes	Immunologic Effect	Protective Effect	Group Recommendation (ACIP, CDC)	Limitation
PCV7 Prevenar	4, 6B, 9V, 14, 18C, 19F, 23F	Stimulate memory B cells through a T dependent mechanism	IPD, pneumonia, otitis media, HIV adults from pneumococcal infection and hospitalization	The same group recommended for PCV13: children < 5 years	Coverage of less serotypes than the others
PCV10 Synflorix	1, 4, 5, 6B, 7F, 9V, 14, 18C, 19F, 23F; eight capsular polysaccharides conjugated to a protein D of non-typeable H. influenzae, and two to tetanus or diphtheria toxoid	Stimulate memory B cells through a T dependent mechanism	IPD through protection from nasopharyngeal colonization	The same group recommended for PCV13: children < 5 years	Coverage of less serotypes than PCV13
PCV13 Prevenar 13	1, 3, 4, 5, 6A, 6B, 7F, 9V, 14, 18C, 19A, 19F, 23F; all are conjugated to CRM197, a non-toxic mutant of diphtheria toxin	Increase serotype-specific memory B-cell responses; induce important T-cell–dependent memory responses; stimulates good antibody response, mucosal immunity, and immunologic memory and systemic anamnestic IgG response in children and adults	Pneumonia in children, and in adults, effectiveness against IPD	Children < 5 years with chronic medical conditions and immunocompromised adults (CSF leaks, cochlear implants, sickle cell disease and other hemoglobinopathies, congenital or acquired asplenia, hematologic malignancies, HIV infection, active cancer, long-term immunosuppressive therapies such as corticosteroids and radiation) [13,16]. Indicated in the first year of life with a series of 2–3 doses in the first semester of life and with a booster dose at one year. Boosters are recommended starting from 6 years for children with severe disease (It is not defined how many PCVs boosters) [14,15]	Coverage of less serotypes than PPV23 High cost
PPV23 Pneumovax 23	1, 2, 3, 4, 5, 6B, 7F, 8, 9N, 9V, 10A, 11A, 12F, 14, 15B, 17F, 18C, 19A, 19F, 20, 22F, 23F, 33F	Promote an immune response determined by the stimulation of B lymphocytes which become activated and then, as plasma cells, produce antibodies. Promote the production of serum IgG but not secretory IGA in the nasopharynx	IPD, seems to alleviate CAP severity	All individuals at increased risk of invasive pneumococcal disease: single dose with a booster about 5 years later, if necessary in patient age > 65, diabetes, nephrotic syndrome, chronic renal failure, CSF leaks, cochlear implants, hemoglobinopathies, congenital or acquired asplenia, hematologic or generalized malignancies, HIV infection, solid organ transplant, primary immunodefciencies, iatrogenic immunodeficiencies.	Poor immunogenicity; Poor effectiveness against pneumococcal pneumonia prevention

ACIP: Advisory Committee on Immunization Practices; CDC: Centers for Disease Control and Prevention.

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
