# Peer review of "Pneumonia and Invasive Pneumococcal Diseases: The Role of Pneumococcal Conjugate Vaccine in the Era of Multi-Drug Resistance"

_vaccines, 2021, doi:10.3390/vaccines9050420_

Round 1

Reviewer 1 Report

In this review, Chiara et al. provide an extensive overview of Streptococcus pneumoniae vaccination strategies available and its well reported effects on incidences of pneumonia and invasive pneumococcal diseases and emergence of serotype replacement and antibiotic resistance. While the review is comprehensive and includes lots of vaccination related statistics, the article lacks coherence and is vague as to the message the authors are trying to deliver through this article. The article fails to explain in detail or cite previous papers regarding how the vaccines work at basic immunology levels, or why PPSV is not as efficient at inciting memory as PCV. Also, given the title of pneumococcal vaccines in era of multidrug resistance, the review does not do adequate justice to describing strategies that could be used to replace or complement the PCV or PPSV vaccines as next generation vaccines (these should include detailed description of various pneumococcal protein vaccines and whole cell vaccines there are being developed and their efficacy in animal/human studies). Overall, the review felt like a lot of statistics and facts but I came out with no take home message. Major restructuring of the article, adding citations for many statements and editing of english in addition to addressing the below mentioned comments will be help greatly improve the paper. Of note, comments below are a small subset of many other in the manuscript and are not exhaustive. Marking line numbers for the article during revision will help the reviewer explicitly mention areas of the document that needs attention.

Citations of the primary literature supporting many of the statements in the review are missing.

Page 2: Please include citation for 5-90% carriage rate.

Page 2: there are now 100 serotypes known. Updating the numbers accordingly and including the respective citations is necessary.

Page 2:  citation #5 doesn’t mention any information regarding the serotype dependent distribution of invasive disease that the authors mention. Please include primary literature to support these numbers.

Page 3: details about what the carrier protein in PCV7 is will be helpful

Page 3: please cite all the primary literature supporting ‘immunologic effects’ section

Page 5: 'pneumolysin'

Page 5: is it 23.2 cases per 100.000 or 23.2 cases per 100,000? Editing numbers to include comma (,) instead of (.) is necessary throughout the document to avoid confusion.

Page 7: “cytokines” spelling needs to be corrected

Page 7: Figure legend needs to consistent with use of words like ‘PCV’ or “CD4+ T cells” or “long lived plasma cells”

Page 8: what does RCT stand for?

Page 8: what does CKD stand for?

Author Response

In this review, Chiara et al. provide an extensive overview of Streptococcus pneumoniae vaccination strategies available and its well reported effects on incidences of pneumonia and invasive pneumococcal diseases and emergence of serotype replacement and antibiotic resistance. While the review is comprehensive and includes lots of vaccination related statistics, the article lacks coherence and is vague as to the message the authors are trying to deliver through this article.

Response: We thank the reviewer for the comment. We are very grateful for the attention given to our manuscript and we tried to carry out the suggested insights as well as possible. In particular, we added some details, we edited the test in a clearer way and we improved the conclusion section to make the message more evident.

The article fails to explain in detail or cite previous papers regarding how the vaccines work at basic immunology levels, or why PPSV is not as efficient at inciting memory as PCV.

Response: We appreciate the reviewer’s comment. We expanded the part dedicated to immunological features of vaccines and we added more references.

Also, given the title of pneumococcal vaccines in era of multidrug resistance, the review does not do adequate justice to describing strategies that could be used to replace or complement the PCV or PPSV vaccines as next generation vaccines (these should include detailed description of various pneumococcal protein vaccines and whole cell vaccines there are being developed and their efficacy in animal/human studies).

Response: We thank the reviewer for the comment. This subject has been covered in more details in the added paragraph entitled “Overview on serotype independent pneumococcal vaccine“ and in the discussion section.

Overall, the review felt like a lot of statistics and facts but I came out with no take home message. Major restructuring of the article, adding citations for many statements and editing of english in addition to addressing the below mentioned comments will be help greatly improve the paper. Of note, comments below are a small subset of many other in the manuscript and are not exhaustive. Marking line numbers for the article during revision will help the reviewer explicitly mention areas of the document that needs attention.

Response: We appreciate the reviewer’s comments. We are aware that the topic is very extensive and it is not easy to deal with all aspects in depth. Our aim was to provide a comprehensive overview highlighting perspectives but also current problems in dealing with the pneumococcal burden. We added various references, we added some chapters and paragraphs to make the review more usable. We also provided a thorough editing of English and we answered to the following comments.

Citations of the primary literature supporting many of the statements in the review are missing.

Response: We thank the reviewer for the comment. We updated the literature and we added citations when necessary.

Here are the point by point answers:

Page 2: Please include citation for 5-90% carriage rate.

Response: We updated the specific reference: Centers for Disease Control and Prevention. Epidemiology and prevention of vaccine-preventable diseases. 13th Edition (2015). Washington DC: Public Health Foundation[7].

Page 2: there are now 100 serotypes known. Updating the numbers accordingly and including the respective citations is necessary.

Response: We updated the total number of serotypes, adding a specific reference: Ganaie, F; Saad, A.S.; McGee, L; van Tonder, A.J.; Bentley, S.D.; Lo, S.W. A New Pneumococcal Capsule Type, 10D, is the 100th Serotype and Has a Large cps Fragment from an Oral Streptococcus. MBio. 2020, 19;11(3): e00937-20. doi: 10.1128/mBio.00937-20 [8].

Page 2:  citation #5 doesn’t mention any information regarding the serotype dependent distribution of invasive disease that the authors mention. Please include primary literature to support these numbers.

Response: We updated a correct reference regarding the serotype dependent distribution of invasive diseases: Oligbu, G; Fry, N.K.; Ladhani, S.N. Chapter 17 The Pneumococcus and Its Critical Role in Public Health in Streptococcus Pneumoniae Methods and Protocols,1st ed.; Iovino, F; Humana Press, New York, NY. 2019; volume 1968, p. 209, https://doi.org/10.1007/978-1-4939-9199-0 [10]

Page 3: details about what the carrier protein in PCV7 is will be helpful.

Response: We added details about the carrier protein in PCV7: the carrier protein induces a T-cell dependent immune response, consisting in immunologic memory and antibody response in adults and young children.

Page 3: please cite all the primary literature supporting ‘immunologic effects’ section.

Response: We updated the references supporting the “immunologic effects” section.

Page 5: 'pneumolysin'.

Response: We corrected this word as requested.

Page 5: is it 23.2 cases per 100.000 or 23.2 cases per 100,000? Editing numbers to include comma (,) instead of (.) is necessary throughout the document to avoid confusion.

Response: We chose to use a comma instead of a dot in numbers where necessary and we corrected them.

Page 7: “cytokines” spelling needs to be corrected.

Response: We corrected this word as requested.

Page 7: Figure legend needs to consistent with use of words like ‘PCV’ or “CD4+ T cells” or “long lived plasma cells”

Response: We tried to better explain the mechanisms of vaccines in the parts of the text related to the two figures. We added extended versions of the words mentioned in the figure legends.

Page 8: what does RCT stand for?

Response: RCT is the abbreviation for “randomized control trial”; we added the extensive form the first time it appears in the text.

Page 8: what does CKD stand for?

Response: CKD is the abbreviation for chronic kidney disease; we added the extensive form the first time it appears in the text.

Reviewer 2 Report

General comment: Please read and revise the manuscript thoroughly and correct the spelling mistakes throughout the manuscript. Some sentences need to be reformatted for the readers to understand them better.

Abstract: 

  1. S. pneumoniae disease ---> Should be either pneumonia or disease caused by S. pneumoniae.
  2. Instead of elder population ---> elderly population
  3. Spread to other parts of the organism ---> other parts of the human host

Correction and comments:

  1. Instead of elder population ---> elderly population, correct throughout the manuscript.
  2. Correct and check the spelling ‘Streptococcus pneumoniae’ throughout the manuscript.
  3. Emergent ---> emerging
  4. 91 different serotypes ---> Please update the number as 100 serotypes have been discovered so far. (Reference: A New Pneumococcal Capsule Type, 10D, is the 100th Serotype and Has a Large cps Fragment from an Oral Streptococcus).
  5. Please correct the spelling of ‘Prevenar’ throughout the manuscript ---> It is Prevnar.
  6. Table 1: Please write the full name for abbreviations ACIP and CDC in the footnote of Table 1.
  7. Page 5: protein pneumolysine ---> Should be pneumolysin.
  8. Page 5: 50% of all causes bacterial meningitis: I think the author meant to say ‘cases’
  9. Main problem ‘regard’ the resistance to ---> Should be ‘regarding the resistance to’
  10. Figure 1: Citochines ---> Cytokines
  11. The discussion section has repetition at some points. Please check it.

Discussion section: Studies on bacterial proteins are ongoing ---> Please elaborate clearly why bacterial proteins ---> as they could provide serotype independent protection

Author Response

General comment: Please read and revise the manuscript thoroughly and correct the spelling mistakes throughout the manuscript. Some sentences need to be reformatted for the readers to understand them better.

Response: We thank the reviewer for the comment. We revised the manuscript thoroughly as requested and we corrected spelling mistakes. We also reformatted some sentences in order to make the text more readable.

Here are the point by point answers:

Abstract:

  1. pneumoniae disease ---> Should be either pneumonia or disease caused by S. pneumoniae.

Response: We rewrote the sentence with the corrections mentioned.

Instead of elder population ---> elderly population. See answer above.

Response: We rewrote the sentence with the corrections mentioned.

Spread to other parts of the organism ---> other parts of the human host.

Response: We rewrote the sentence with the corrections mentioned.

Correction and comments:

Instead of elder population ---> elderly population, correct throughout the manuscript.

Response: We corrected the word throughout the manuscript as requested.

Correct and check the spelling ‘Streptococcus pneumoniae’ throughout the manuscript.

Response: We corrected the word throughout the manuscript as requested and after the first use we continued with the acronym SP.

Emergent ---> emerging.

Response: We corrected the word throughout the manuscript as requested.

91 different serotypes ---> Please update the number as 100 serotypes have been discovered so far. (Reference: A New Pneumococcal Capsule Type, 10D, is the 100th Serotype and Has a Large cps Fragment from an Oral Streptococcus).

Response: We updated the number and we added the suggested reference as requested.

Please correct the spelling of ‘Prevenar’ throughout the manuscript ---> It is Prevnar.

Response: We corrected the word throughout the manuscript as requested.

Table 1: Please write the full name for abbreviations ACIP and CDC in the footnote of Table 1.

Response: We wrote the full names for the abbreviations mentioned in the footnote of Table 1.

Page 5: protein pneumolysine ---> Should be pneumolysin.

Response: We corrected the word throughout the manuscript as requested.

Page 5: 50% of all causes bacterial meningitis: I think the author meant to say ‘cases’.

Response: We corrected the word on page 5 as requested; “cases” is the correct one.

Main problem ‘regard’ the resistance to ---> Should be ‘regarding the resistance to’.

Response: We corrected the verb as requested.

Figure 1: Citochines ---> Cytokines.

Response: We corrected the word in the figure as requested.

The discussion section has repetition at some points. Please check it.

Response: We revised the discussion section and we simplified it by removing repetitions and separating the several points of discussion.

Discussion section: Studies on bacterial proteins are ongoing ---> Please elaborate clearly why bacterial proteins ---> as they could provide serotype independent protection.

Response: We rewrote the sentence expanding the concept mentioned. Indeed, strategies that could be used to replace or complement PCV or PPV as next generation vaccines are added in a new paragraph named “Overview on serotype independent pneumococcal vaccine” and in the discussion section.

Reviewer 3 Report

Thank you for sending me the research article paper “Pneumonia and invasive pneumococcal diseases: the role of pneumococcal conjugate vaccine in the era of multidrug resistance” for review in the Vaccines. In the article of Chiara et al., the author discussed the role of pneumococcal conjugate vaccine in the era of multi-drug resistance. There are important points that should be discussed and improved.

  1. Author should discuss the role and mechanism of Streptococcus pneumoniae in the development of diseases. Why is this infectious agent dangerous for the diseases? Discuss in detail along with the type of diseases especially the role of bacteremia.
  2. Epidemiological data should present in different regions of the world to explain the importance of infection.
  3. Clinical features of invasive and non-invasive infection: author should divide into several sub heading to understand more easily.
  4. Therapeutic option: author should divide into several sub heading to understand more easily.
  5. Author should add more paragraphs regarding the development stages of the vaccine. How successful are present vaccines.
  6. Author should explain the recent development of pneumococcal vaccine and clinical trial.
  7. Author should discuss the failed attempt of the development of vaccines and their reasons.
  8. Author should include more graphical representation of the mechanism of vaccines.

Author Response

Comments and Suggestions for Authors

Thank you for sending me the research article paper “Pneumonia and invasive pneumococcal diseases: the role of pneumococcal conjugate vaccine in the era of multidrug resistance” for review in the Vaccines. In the article of Chiara et al., the author discussed the role of pneumococcal conjugate vaccine in the era of multi-drug resistance. There are important points that should be discussed and improved.

Response:  We thank the reviewer for the comments. We are very grateful for the attention given to our manuscript and we tried to carry out the suggested insights as well as possible.

Here are the point by point answers:

Author should discuss the role and mechanism of Streptococcus pneumoniae in the development of diseases. Why is this infectious agent dangerous for the diseases? Discuss in detail along with the type of diseases especially the role of bacteremia.

Response: We added specific information on the mechanism of Streptococcus pneumoniae in the dedicated paragraph: “Streptococcus pneumoniae serotypes and the mechanism of infection”.

Epidemiological data should present in different regions of the world to explain the importance of infection.

Response:  We updated data about the epidemiology in the introduction section.

Clinical features of invasive and non-invasive infection: author should divide into several sub heading to understand more easily.

Response: We divided paragraphs and we added subheadings in order to better recognize them and make them more readable.

Therapeutic option: author should divide into several sub heading to understand more easily.

Response: We divided paragraphs and we added subheadings in order to better recognize them and make them more readable.

Author should add more paragraphs regarding the development stages of the vaccine. How successful are present vaccines.

Response: We revised the manuscript actually finding several sentences describing how present vaccines are successful, therefore we decided not to further expand it, as the manuscript is already extended. We highlight here some key sentences, for example: ” Routine use of PCV in infants leads to an unexpected reduction in nasopharyngeal colonization with vaccine serotypes of SP [1]. As children are the main reservoir and transmitters for SP diseases in adults, the reduction in serotypes as colonizers of infants is an important factor in increasing of herd immunity.”

However, more details have been added regarding the development history of current vaccines. Such as: “The first is a preparation containing 23 serotypes (PPV23), based on capsular polysaccharides, and it was licenced in 1983 in the Unted States (US), replacing the first pneumococcal vaccine approved in 1977 with 14 serotypes [7,14]. One dose of PPV23 is currently recommended by Centers for disease control and prevention (CDC) for adults aged >65 years in the US, eventually preceded by a dose of PCV13 one year before in case of increased risk for exposure to PCV13 serotypes; the combination is as well recommended for adults aged ≥19 years with immunocompromising condition, CSF leak, cochlear implant [14].

Author should explain the recent development of pneumococcal vaccine and clinical trial.

Response: we added a new paragraph in the conclusion section in order to summarize recent development of pneumococcal vaccines and trials.

Author should discuss the failed attempt of the development of vaccines and their reasons.

Response: We pointed out several problems in the current vaccination strategies in the discussion section, highlighting possible perspectives. We are aware that the topic is very extensive and it is not easy to deal with all aspects in depth.

Author should include more graphical representation of the mechanism of vaccines.

Response: We added another figure showing the mechanism of PPV23 compared to the PCV one.

Reviewer 4 Report

I was invited to revise the paper entitled "Pneumonia and invasive pneumococcal diseases: the role of pneumococcal conjugate vaccine in the era of multi-drug resistance". It was a review that aimed to provide an overview on Streptococcus pneumonia and the clinical features of Streptococcus pneumonia related diseases.

The topic is relevant for public health and focused on impacting strategies for diseases prevention.

The review is well written and easy to read.

In my opinion Authors missed some points that need to be addressed:

  • Authors have to specify how different countries manage the PC vaccination strategies. In particular, Italy for older adults suggest PC 13conjugate vaccine prior 23polysaccaridic vaccine. Some country, as Spain start directly with 23valent vaccine;. It can be desumed from "Immunological features" section, but Authors should specify and comapare different vaccination strategies;
  • About costs and outcomes, Authors should focus also on PC related hospitalization;
  • About AMR, Authors cannot limit their discussion to the impact of vaccination on resistance. They should highlight the background of AMR in Europe and other Continent, comparing also differences in antibiotic treatment.

Author Response

I was invited to revise the paper entitled "Pneumonia and invasive pneumococcal diseases: the role of pneumococcal conjugate vaccine in the era of multi-drug resistance". It was a review that aimed to provide an overview on Streptococcus pneumonia and the clinical features of Streptococcus pneumonia related diseases.

The topic is relevant for public health and focused on impacting strategies for diseases prevention.

The review is well written and easy to read.

Response:  We thank the reviewer for the comments. We are very grateful for the attention given to our manuscript and we tried to carry out the suggested insights as well as possible.

Here are the point by point answers:

In my opinion Authors missed some points that need to be addressed:

Authors have to specify how different countries manage the PC vaccination strategies. In particular, Italy for older adults suggest PC 13conjugate vaccine prior 23polysaccaridic vaccine. Some country, as Spain start directly with 23valent vaccine;. It can be desumed from "Immunological features" section, but Authors should specify and compare different vaccination strategies;

Response: We expanded the information about US strategies and we added specific information about UK vaccination strategy to the ones already provided for Italy.  

About costs and outcomes, Authors should focus also on PC related hospitalization;

Response: We added a specific paragraph with the correlated references about PC related hospitalization.

About AMR, Authors cannot limit their discussion to the impact of vaccination on resistance. They should highlight the background of AMR in Europe and other Continent, comparing also differences in antibiotic treatment.

Response: We added a dedicated paragraph to explain differences in antibiotic treatments in Europe and other Continents

Round 2

Reviewer 1 Report

I have only one minor concern with the figure legend of Fig 2.

It should be 

"Figure 2. Schematic representation of the T-dependent mechanism of the PCV 13 vaccine. The polysaccharidic portion of PCV stimulates the production of plasma cells, while the carrier portion of PCV activates CD4+ T lymphocytes that induces memory B cell proliferation and long lived Plasma cells. "

Author Response

We thank the reviewer for the comment. We reformatted the sentence.

Reviewer 3 Report

NA

Author Response

We thank the reviewer for the attention given to our manuscript.

Reviewer 4 Report

Paper is now acceptable for publication

Author Response

(The authors gave the same response as above.)
